# Functional Dissection of the Physiological Traits Promoting Durum Wheat (*Triticum durum* Desf.) Tolerance to Drought Stress

**DOI:** 10.3390/plants12071420

**Published:** 2023-03-23

**Authors:** Salim Ltaief, Abdelmajid Krouma

**Affiliations:** 1Faculty of Sciences of Gafsa, Sidi Ahmed Zarroug, Gafsa 2112, Tunisia; 2Faculty of Sciences and Techniques, Sidi Bouzid 9100, Tunisia; 3Faculty of Sciences of Sfax, Road la Soukra km 4-BP, Sfax 1171-3000, Tunisia

**Keywords:** drought susceptible index, drought tolerance index, durum wheat, photosynthesis, proline, relative osmolyte content

## Abstract

In Tunisia’s arid and semi-arid lands, drought stress remains the most critical factor limiting agricultural production due to low and irregular precipitation. The situation is even more difficult because of the scarcity of underground water for irrigation and the climate change that has intensified and expanded the aridity. One of the most efficient and sustainable approaches to mitigating drought stress is exploring genotypic variability to screen tolerant genotypes and identify useful tolerance traits. To this end, six Tunisian wheat genotypes (*Triticum durum* Desf.) were cultivated in the field, under a greenhouse and natural light, to be studied for their differential tolerance to drought stress. Many morpho-physiological and biochemical traits were analyzed, and interrelationships were established. Depending on the genotypes, drought stress significantly decreased plant growth, chlorophyll biosynthesis, and photosynthesis; stimulated osmolyte accumulation and disturbed water relations. The most tolerant genotypes (salim and karim) accumulated more potassium (K) and proline in their shoots, allowing them to maintain better tissue hydration and physiological functioning. The osmotic adjustment (OA), in which potassium and proline play a key role, determines wheat tolerance to drought stress. The calculated drought index (DI), drought susceptible index (DSI), drought tolerance index (DTI), K use efficiency (KUE), and water use efficiency (WUE) discriminated the studied genotypes and confirmed the relative tolerance of salim and karim.

## 1. Introduction

Water stress has become a global problem due to climate change. The Intergovernmental Panel on Climate Change reported drought as the main environmental factor threatening crop productivity worldwide [1]. In the Mediterranean region, drought remains the most critical factor limiting agricultural production and can occur at any time of the plant cycle due to low and irregular precipitation. As throughout the world, global climate change has accentuated the frequency and severity of drought stress in Tunisia, and cultivated areas are expected to decrease shortly [2,3].

Wheat crops occupy about 218 million acres and account for 1/3 of the world’s cereal production, with 771 million tons of production per year that satisfies the demand of 21% to 36% of the world’s population. Mustapha et al. [4] recommend increasing wheat yield by up to 60% to feed the world population in 2050. However, wheat is known to be sensitive to drought stress, particularly during the critical stage of the plant cycle, such as ear and grain filling, resulting in a 58–92% yield reduction [5].

For adaptive morphological traits, Skirycz et al. [6] demonstrated that various adaptive mechanisms control plant survival and development in water deficit conditions. In durum wheat, tolerant varieties develop fewer roots in the topsoil when dense roots develop deeper [7,8,9]. Root elongation in deep soils is an essential adaptive trait [10,11]. Some drought effects on crops are irreversible, such as leaf morphology, reduced leaf number, and increased root-to-shoot ratio [12,13]. Reduced leaf area is also another trait resulting from the restricted transfer of sugars due to hampered photosynthesis [14].

The physiological parameters related to plant response to drought stress at the vegetative phase implied reduced water content and water potential [15,16,17], stomatal closure [18,19], and osmotic adjustment [20]. The latter involves compatible solute accumulation in the cells to decrease the osmotic potential such as proline [17,21], sugars [22,23], and other organic and inorganic solutes [24]. This mechanism allows the maintenance of turgor for as long as possible. Maqbool et al. [25] reported that drought tolerance depends primarily on OA to maintain cell turgor and delay dehydration. Such a situation helps maintain the vital physiological processes of the plant. Ruszkowski et al. [26] demonstrated that soluble sugars and proline are the main substances of OA in wheat crops. Maqbool et al. [25] reported that proline content is an essential criterion for selecting tolerant genotypes. Multiple studies have demonstrated that drought stress increases the proline content in leaves during the vegetative and reproductive stages of the plant cycle [25].

Similarly, the accumulation of soluble sugars has been shown to play a key role in decreasing the water potential and osmotic adjustment in different durum wheat genotypes, giving them some ability to tolerate water stress [27]. Sugars are also implied in the maintenance of phosphorylation, energy production, and the protection of enzyme biosynthesis [27]. Abdalla [28] reported that the accumulation of sugars is a simple drought adaptation mechanism, which gives the plant the ability to maintain its turgor through decreasing water potential. Otherwise, numerous studies have demonstrated the crucial role of K in osmoregulation under osmotic stress (drought and salinity). It is well documented that an adequate K supply enhances crop production and resistance to numerous abiotic and biotic stresses [29,30,31]. Britto et al. [32] demonstrated that maintaining high K concentrations in salt-tolerant genotypes could be one of the mechanisms underlying their high salt tolerance. Tränkner et al. [30] reported that K plays a crucial role in stress mitigation. Tavakol et al. [33] demonstrated that K is the most abundant cation in plants, playing an essential role in osmoregulation. 

Durum wheat is a strategic crop in Tunisia, cultivated in winter and harvested in summer. It occupies most of the agricultural land in the north, where rainfall usually meets the water needs of plants. The main critical stage of the plant cycle is the vegetative development that influences the reproductive stage (flowering, ear setting, and ear filling). However, most of the research on drought stress has been conducted at the grain-filling stage, with little attention given to the vegetative stage. Furthermore, climate change has pushed aridity to spread over almost all of the Tunisian land and durum wheat fields (with specific rain-fed genotypes) that were in humid climates now belong to the semi-arid or arid climate. Their response to water scarcity must therefore be studied. Most of the research on drought stress in wheat is done at the grain-filling stage, also known as terminal drought or postanthesis drought [34]. Exploring the genotypic variability of response to drought stress among wheat genotypes allows us to screen for tolerant ones and identify the associated useful traits of tolerance. Accordingly, this study was planned and consists of subjecting six Tunisian genotypes to water stress (karim, khiar, inrat, maali, razek, and salim). They are widely cultivated in the north of the country, where the climate is generally humid, while karim crops extend to the center (semi-arid climate). A particular interest will be granted to the key metabolic reactions such as chlorophyll biosynthesis, photosynthesis, K nutrition, water relations, and osmolyte accumulation with respect to their interrelationships.

## 2. Results

The daily monitoring of plant morphology showed a slight shoot yellowing of plants subjected to drought stress, representing chlorophyll degradation. The Spad index significantly decreased under drought stress (Table 1). Depending on genotypes, the spad index decreased by 26, 28, 35, 40, 41, and 44%, respectively, in salim, karim, razek, khiar, inrart100, and maali, as compared to control plants. In the same way, the quantification of biomass production demonstrated the harmful effect of water scarcity on plant growth. Plant biomass decreased by 28% in salim and karim and exceeded 40% in the other genotypes (44% in maali, 50% in khiar, 51% in razek, and 53% in inrat, as compared to control plants, Table 1). However, even if decreased, the plant biomass produced by salim and karim exceeded that of maali and inrat by more than 50%. Thus, a general behavior exists among the studied genotypes with a clear difference in response to drought stress. 

To progress in this investigation, we measured gas exchange parameters. Figure 1a shows that the exposure of wheat plants to drought stress dramatically hampers net photosynthesis in all genotypes. Nevertheless, the previously observed behavior was maintained, and genotypes that demonstrated better chlorophyll content and plant growth maintained their superiority in photosynthetic activity. Net photosynthesis decreased by 35 to 38% in razek, karim, and salim and by 54 to 66% in maali, inrat, and khiar (Figure 1a). High values of An under drought stress are observed in karim and salim. These genotypes developed net photosynthetic activity 1.4 to 2 times higher than maali, inrat, and khiar. For the stomatal conductance, Figure 1b shows the same behavior as net photosynthesis, particularly under drought stress where karim and salim are discriminated from maali, inrart, and khiar. Genotypes that have developed better net photosynthesis express higher stomatal conductance. The same general behavior was also observed for evapotranspiration (Figure 1c). 

To study the repercussions of drought stress on water relations, we measured the first-time water potential (ψw). Figure 2a demonstrates that drought stress significantly increased the negativity of this parameter. Nevertheless, genotypes that have maintained better shoot hydration (salim and karim) have demonstrated a high ability to decrease their water potential. Water potential decreased by 42%, 67%, 53%, 28%, 24%, and 26% in razek, salim, karim, maali, inrart, and khiar, respectively, compared to control plants. A second time, we calculated the relative water content (RWC). Figure 2b shows that RWC decreased in leaves subjected to water deficits, depending on genotype. This decrease was significant in maali (−22%), inrat (−20%), khiar (−21%), and razek (−12%) but not significant in salim (−4%) and karim (−8%).

It is well established that some crop plants respond to drought stress by accumulating mineral and/or organic osmolytes to adjust their water potential. For this purpose, we analyzed the relative osmolyte content (ROC) in leaves. Obtained results demonstrated that ROC significantly increased under drought stress in all genotypes. This increase was estimated at 1%, 12%, 17%, 22%, 39%, and 59%, respectively, in khiar, inrat, maali, razek, karim, and salim (Figure 3a). Figure 3b, which illustrates proline content, showed a clear stimulation of this osmolyte in shoots of durum wheat subjected to water scarcity. The proline concentration was 3.4 to 5.3 times higher in stressed plants as compared to control ones (3.4, 4.2, 4.3, 4.5, 5.3, and 5.3 times, respectively, in khiar, inrat, razek, maali, karim, and salim). Regarding these results, salim and karim maintained their high capacities of OA as compared to the other genotypes.

The analysis made on potassium nutrition demonstrated that when subjected to drought stress, K concentration increased in shoots of salim and karim while decreasing in the other genotypes. In roots, no clear modification was observed (Figure 4). The shoot/root ratio calculation demonstrated an exceptional increase in salim and karim under drought, reflecting an important uptake and allocation of K to shoots (Table 1). In light of these significant results, we calculated several parameters to express the genotypic differences observed in this study. Drought index calculated based on the Spad index (DI-Spad), biomass production (DI-DW), or net photosynthesis (DI-An) showed the lowest values in the genotypes that demonstrated higher plant growth, photosynthetic activity, and K accumulation in shoots (salim and karim). In contrast, the highest values reflecting higher sensitivity are observed in the genotypes expressing the lowest biomass production, photosynthesis, and K concentrations (inrat, maali, and khiar) (Table 1). DI-Spad did not exceed 0.26 and 0.28, respectively, in karim and salim, when reaching 0.35, 0.40, 0.41, and 0.44, respectively, in inrat, razek, khiar, and maali. While respecting the same arrangement of genotypes, DI-DW was about 0.29 and DI-An was about 0.37 in karim and salim when exceeding 0.5 in the other genotypes (Table 1). In addition to these physiological traits, we calculated K use efficiency for plant growth (KUE-DW), K use efficiency for photosynthesis (KUE-An), water use efficiency for plant growth (WUE-DW), and water use efficiency for photosynthesis (WUE-An). Table 2 shows that, independently of the parameter used for calculation, KUE decreased significantly in plants subjected to drought stress compared to control plants. However, the tolerant genotypes, salim and karim, expressed the highest efficiencies of K use when subjected to water scarcity. The calculated WUE also discriminated against the studied genotypes and confirmed the relative tolerance of salim and karim compared to the other genotypes (Table 2). WUE-DW decreased by 44%, 41%, 37%, 28%, 26%, and 22%, respectively, in razek, inrart, khiar, maali, salim, and karim. For WUE-An, drought stress decreased this parameter significantly in all genotypes (−56%, −49%, −43%, −37%, −33%, and −33%, respectively, in maali, khiar, inrart, razek, salim, and karim).

Drought susceptible index (DSI) and drought tolerance index (DTI) are demonstrated useful traits to highlight the genotypic variability of response to drought stress. For this purpose, we calculated these two traits based on spad, photosynthesis, and plant growth (Table 3). The obtained results confirmed the relative tolerance of salim and karim genotypes as they expressed the lowest values of DSI (DSI-spad, DSI-An, and DSI-DW) and the highest values of DTI (DTI-spad, DTI-An, and DTI-DW).

To progress in elucidating the physiological mechanisms promoting the genotypic differences in the response of durum wheat to drought stress, we established several correlations between the studied physiological parameters. The correlation between plant growth and drought index (Figure 5a) and between net photosynthesis and drought index (Figure 5b) showed a strict negative relationship in plants subjected to drought stress (R^2^ = 0.795). Genotypes that expressed better biomass accumulation and photosynthetic activity (salim and karim) showed the lowest values of the drought index and, inversely, for the other genotypes. Nevertheless, these relationships lacked in control plants.

Obtained results on ROC, proline, and K repartition gave us a clear indication of their implication in the response of durum wheat to drought stress and the genotypic differences in the tolerance to this osmotic stress. For that, we correlated DI-DW with ROC (Figure 6a), with proline (Figure 6b), and with K concentration (Figure 6c). Figure 6 shows a close, strict, and negative relationship between the drought index and these osmolytes. Being genotype dependent, increasing ROC, proline, and/or K accumulation decreased DI-DW. 

In a second time, we correlated relative water content (RWC) with the same above osmolytes. Figure 7 shows a close and strict, but positive, relationship between RWC and ROC (a), between RWC and proline (b), and between RWC and K (c) when plants are subjected to drought stress. Genotypic differences are also maintained; salim and karim expressed higher potentialities for osmolyte accumulation and maintained better shoot hydration. In control plants, we noticed a lack of clear relationships between these parameters (not shown).

To confirm the interdependence of shoot hydration and drought tolerance and the respective genotypic differences, we correlated RWC with DI-DW (Figure 8). Obtained results demonstrated the close negative relationship between these two physiological parameters. The genotype that maintains better shoot hydration under drought stress shows the lowest drought index, reflecting its relative tolerance (salim and karim in this study).

Finally, we correlated biomass production with shoot K concentrations (Figure 9a) and net photosynthesis with shoot K concentrations (Figure 9b). Even though it was lacking in control plants (not shown), a close positive relationship was established between the key physiological parameters (plant growth and photosynthesis) and K nutrition in drought-stressed plants. The most tolerant genotypes (salim and karim) showed the highest plant growth, photosynthetic activity, and shoot K concentration. A reverse behavior was observed in the most sensitive genotype (inrat) and in the intermediate positions for the other genotypes.

## 3. Discussion

Drought is an abiotic stress that occurs when plants are subjected to prolonged water deficit. That stress could have various origins (atmospheric due to low precipitation, agricultural linked to irrigation systems, soil because of surface and underground water scarcity, and physiological resulting from high transpiration) [35]. At the vegetative stage, drought affects the crucial metabolic processes, which have a repercussion on the final plant growth.

In the present study, we investigated the mechanisms underlying wheat tolerance to drought stress and the correlation of morpho-physiological and biochemical traits. The exposure of durum wheat to a water deficit inhibits the spad index and plant growth with some genotypic differences. Salim and karim were found to be relatively tolerant compared to razek, inrart, khiar, and maali. These two genotypes produced 1.3 to 2 times more biomass under water stress than the others and showed high photosynthetic assimilation. The calculation of the DI, DSI, and DTI discriminated clearly between the studied wheat genotypes and confirmed the relative tolerance of salim and karim. In fact, it is well reported that plants could tolerate drought stress by changing their physiological functions, such as less reduction in chlorophyll and water content, biomass production, membrane stability, photosynthetic activity; higher accumulation of soluble sugars, proline, and amino acids; and stimulated enzymatic and non-enzymatic activities [36]. Mohammadian et al. [37] observed a significant decrease in leaf area and biomass production when sugar beet genotypes were subjected to water stress. In Okra plants, drought significantly reduced plant size and decreased cell elongation [38]. Singh et al. [39] reported a significant decrease in biomass production, plant size, and seed yield in chickpea germplasms subjected to drought stress. In the same species, Krouma [40] highlighted genotypic differences in response to drought stress and identified the most tolerant genotype (Amdoun), which expressed the lowest DI (0.36). Rosales-Serna et al. [41] obtained similar results in common beans subjected to water stress with some genotypic differences; yield reduction is more severe in the sensitive genotypes than in the tolerant ones. Ping et al. [42] demonstrated that increasing water stress decreased soil water content in parallel with decreased leaf hydration, net assimilation, and stomatal conductance. Jia et al. [43] reported that drought stress damaged the reaction centers of PSI and PSII and decreased the efficiency of electron transfer. Blaya-Ros et al. [44] observed stomatal regulation and lower plant growth under water stress. Other authors [45] have shown that photosynthetic limitation develops gradually and is accompanied by PSII failure. Recovery begins immediately after rewatering. Photosynthetic activity and water parameters decreased by more than 80%, associated with leaf water potential declining below −3.0 MPa [46]. According to Long et al. [47], a strict relationship exists between intracellular CO_2_ concentration and CO_2_ assimilation. In the present study, we suggest that the reduced photosynthesis measured under water stress is closely related to the decreased water potential, which depends on the stomatal closure. Such disturbances would result in a clear restriction of CO_2_ diffusion and limited CO_2_ fixation in the Calvin cycle. Jia et al. [42] demonstrated that under drought stress, plants limit evapotranspiration by reducing the opening of stomata. This regulation may cause a restriction of CO_2_ availability for photosynthesis and a decrease in primary photochemical processes. Under these conditions, several processes may contribute to the maintenance of electron flow, such as the Mehler reaction and the photorespiratory process [48]. Otherwise, our measurements of water relations and osmolyte accumulation have shown that water stress decreased RWC and ψw, but increased ROC and proline in shoots. The identified tolerant genotypes (karim and salim) maintained better leaf hydration and accumulated more potassium, osmolyte, and proline. The use of RWC as an indicator of plant water status is important and useful. RWC in leaves is responsive to drought stress and correlates with drought tolerance. It is a better indicator of drought stress than any indices of plants [36]. In *Lupinus albus*, Pinheiro et al. [49] observed a strict relationship between RWC and ψw during progressive water deficit. They supported the use of RWC as an indicator of plant water status. Khakwani et al. [50] demonstrated that wheat genotypes with reduced leaf water loss due to drought stress are believed to be more drought tolerant. RWC may be used as a useful indicator to screen out wheat genotypes with superior drought tolerance. In the present study, salim and karim genotypes maintained better leaf hydration than the other genotypes. Their high ability to preserve leaf turgor and maintain better leaf hydration and photosynthetic activity let us think that these genotypes expressed higher capacities of osmotic adjustment through the accumulation of organic and inorganic compounds. Indeed, osmotic adjustments allow for lowering the cell osmotic potential through the accumulation of osmolytes under drought stress [51], delaying the drought-responsive stomatal closure, maintaining cell turgor, sustaining photosynthesis, and related metabolic functions, and consequently, the maintenance of almost normal growth and development activities [52]. As demonstrated by others [30,53,54], the present study highlighted the close relationship between high biomass yield, improved net photosynthesis, sufficient K availability, and organic osmolytes (proline) accumulation under drought stress, particularly in the tolerant genotypes salim and karim. In fact, the analyzed ROC, proline, and K repartition and relationships in this study confirmed the implication of these components in the relative tolerance of salim and karim compared to the other genotypes. In fact, Figure 6 shows a strict relationship between DI and ROC, proline, and K, with a special behavior of salim and karim, which are shown to be the most accumulators of these components and the least stressed. Thus, we suggest that the high accumulation of proline and K in shoots participates in maintaining leaf hydration, key metabolic reactions (photosynthesis …), and plant growth. This suggestion is confirmed by the results of Tavakol et al. [33], who reported that K is the most abundant cation in plants, playing an important role in osmoregulation. When drought stress occurs, the availability of plant nutrients is decisive in plant stress adaptation and avoidance. Potassium is the most abundant cation in plants and affects the osmotic potential of plant cells, consequently playing a crucial role in stress mitigation [30]. Adequate K availability under osmotic stress improves plant water uptake, and water use efficiency decreases the ABA content, and avoids limitations in stomatal conductance and assimilation [33,55,56]. Otherwise, Chowdhury et al. [36] reported higher capacities of proline accumulation in the tolerant genotypes of wheat, BAW1158 and BAW 1169, subjected to drought stress. Mwadzingeni et al. [21] demonstrated in bread wheat that high proline levels enable plants to attain a low water potential and thus impart tolerance against moisture deficiency by increasing the biosynthesis of intermediate enzymes. Abdalla [28] reported that the accumulation of sugars is a simple drought adaptation mechanism that gives the plant the ability to maintain its turgor through decreasing water potential. Similar studies have reported a positive and strict correlation between growth and osmotic adjustment in pea plants under water stress [57]. Accordingly, the present results confirmed the close relationship between the RWC and ROC, proline, and K (Figure 7), and between the RWC and DI (Figure 8), and discriminated salim and karim from the other genotypes as the most tolerant ones. Otherwise, several correlations have been established between plant growth and DI (negative), photosynthesis and DI (negative), plant growth and shoot K concentration (positive), and between photosynthesis and shoot K concentration (positive). Thus, a common thread seems to exist in the wheat plants under drought stress, which makes the link between all these physiological and biochemical traits and can explain, to some extent, the relative tolerance of salim and karim compared to the other genotypes. Plant growth is highly dependent on tissue hydration and photosynthesis. Genotypes that maintained high shoot hydration under drought stress tend to have higher capacities for preserving water balance in shoots and the respective cytoplasmic metabolic reactions. The relative tolerance of salim and karim genotypes seems to be the result of a good balance between shoot hydration, osmotic adjustment, photosynthesis, and plant growth. 

By progressing in elucidating the physiological traits of wheat tolerance to drought stress and analyzing KUE, WUE, DSI, and DTI, the previously observed genotypic differences are confirmed. Other traits of tolerance to drought stress are thus identified. The WUE and KUE increased gradually with the tolerance of wheat to water scarcity (Table 1 and Table 2). The calculated drought index did not escape this rule and confirmed the observed genotypic differences in the response of durum wheat to drought stress. DSI and DTI are other traits of drought tolerance that discriminated the studied genotypes. The most tolerant genotypes, salim and karim, are less vulnerable to water stress because of their highest DTI and lowest DSI. 

Finally, our results have demonstrated that plant growth, photosynthesis, and osmotic adjustment are interdependent. These relationships become stricter under drought stress and express the genotypic differences. The tolerance of durum wheat to drought stress depends entirely on its capacity of OA, in which K and proline play a key role. Salim and karim’s relative tolerance results from a good balance between several physiological and biochemical functions, including OA, KUE, and WUE, photosynthesis, and stomatal conductance. Their capacity of OA through K and proline accumulation allows them to decrease the osmotic potential and maintain cell turgor for an almost-normal metabolic functioning, whereas their KUE and WUE allow them to use efficiently these minerals for economic metabolic functioning. KUE, WUE, DI, DSI, and DTI are useful physiological traits of drought tolerance that can be used for further screening programs. Using these traits, the studied durum wheat genotypes can be classified according to their degree of tolerance, from the most tolerant to the most sensitive, as follows:
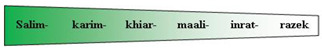


## 4. Materials and Methods

### 4.1. Plant Material and Experimental Design

The present work focused on the strategic crop durum wheat (*Triticum durum* Desf.) in response to drought stress. Six Tunisian genotypes originated from the National Institute of Field Crops (Kef, Tunisia) were used: karim, salim, maali, razek, khiar, and inrart. Previously selected grains for their health and size uniformity are sown directly in the soil under the greenhouse to avoid rains that can disrupt our treatments, without organic or chemical fertilization. The experiment was carried out in the experimental plot of the Faculty of Sciences and Techniques of Sidi Bouzid (35°2′7.58″ N, 9°29′2.18″ E) under natural light, at a temperature of 25 °C/17 °C (±2 °C, day/night), with a relative humidity of about 75%. Soil humidity was maintained at field capacity during the first 4 weeks. Then, homogenous plantlets from each genotype (100 plantlets per plot of 2 m × 1 m dimensions) were maintained, and drought stress was applied. The experimental design was two factorials arranged in a completely randomized design with 3 replications (3 control plots containing 100 plants each irrigated at field capacity, control, and 3 stressed plots containing 100 plants each without irrigation, stressed). After 3 weeks of drought stress (the daily monitored relative soil moisture, RSM, reached 30%), nondestructive measurements (water potential, spad index, and gas exchange) were carried out. Ten plants per plot for a total of thirty plants per treatment were harvested for fresh and dry weight determination and subsequent analysis.

### 4.2. Nondestructive Measurements

#### 4.2.1. SPAD Index

Shoot chlorophyll content was estimated in vivo using a SPAD-502 Unit (Konica-Minolta, Chiyoda-Tokyo, Japan, manufactured by Hangzhou Mindfull Technology Co., Ltd., Hangzhou, China) before gas exchange measurements on one fully expanded leaf. Ten plants per genotype and plot for a total of thirty plants per treatment were analyzed. Results are presented as the mean of 30 replicates, and values are expressed as SPAD units.

#### 4.2.2. Gas Exchange

Gas exchange measurements were conducted with an LI-6400 (LI-COR, Inc., Lincoln, NE, USA) portable gas exchange system. Measurements were carried out on the same 30 plants used for Spad measurements for each genotype and treatment. Photosynthesis was induced with saturating light (1000 μmol m^−2^ s^−1^). This light was fitted to the standard 6 cm^2^ clamp on the leaf chamber. The sample pCO_2_, flow rate, and temperature were kept constant at 362 mbar, 500 μmol s^−1^, and 25 °C, respectively.

#### 4.2.3. Water Potential

Leaf water potential (ψw) was measured according to Scholander et al. [58] on the upper, fully expanded leaves 2 h after sunrise using a pressure chamber (model C52-SF, WESCOR, Inc., South Logan, UT, USA). Measurements are made on ten plants.

### 4.3. Proline Measurements

The proline content of control and stressed wheat leaves was estimated at the final harvest following the standard method [59]. Briefly, 0.5 g of fresh weight from each plant (10 plants per genotype and per treatment) was used for proline estimation. At first, the ninhydrin reagent was prepared as follows: 30 mL glacial acetic acid and 20 mL 6 M orthophosphoric acid were mixed with 1.25 g of ninhydrin. Proline standards (0, 2, 4, 6, 8, 10, 12, 14, 16, 18, and 20 µg/mL) were prepared with deionized water. Using a mortar and pestle, 0.5 g of fresh sample was ground and thoroughly homogenized in 3% sulpho salicylic acid (10 mL) until digestion of plant material was complete. The filtration of homogenate was performed using filter paper (Whatman N°. 2). Then, in a hermetically sealed tube, filtrate (2 mL) and standard proline solution were reacted with ninhydrin reagent (2 mL) and glacial acetic acid (2 mL). These were subsequently boiled in a water bath for 1 h at 100 °C. Subsequently, cooling of the mixture in an ice bath was performed, and toluene (4 mL) was added to each tube. An electrical shaker was used to shake the tubes for 15–20 s to allow the layers to separate. The spectrophotometer (UV/VIS, UV-3100 PC/VWR) at 520 nm, with pure toluene as a blank, was used for the absorbance of the layer. From a standard curve, proline content was estimated on a fresh weight basis for ten plants per genotype and treatment.

### 4.4. Potassium Analysis in Plant Tissues

Samples of fresh material were dried at 70 °C for 72 h and ground to a fine powder. After extraction in 0.5% HNO_3_, ions were measured using a flame photometer (JENWAY, model PFP7, Chelmsford, UK).

### 4.5. Criteria and Parameters of Analysis

Numerous parameters are calculated:

* Drought index, which reflects the severity of the drought stress, was calculated based on the measured parameter (MP) in control and stressed plants according to Fischer and Maurer [60] as follows:$$DI=1-\frac{MPs}{MPc}$$

MP was the Spad index when DI was measured based on Spad (DI-Spad).

MP was dry weight when DI was measured based on plant growth (DI-DW).

MP was net photosynthesis when DI was measured based on photosynthesis (DI-An).

* Drought susceptible index (DSI) was calculated according to Fischer and Maurer [58], whereas the drought tolerance index (DTI) was calculated according to Goudarzi and Pakniyat’s [61] methods adapted to drought stress, as follows:$$DSI=\left[\frac{\left[1-\frac{DWs}{DWc}\right]}{\left[1-\frac{\bar{DW}s}{\bar{DW}c}\right]}\right]\;DTI=\left[\frac{DWs\times DWc}{{\left(\bar{DWc}\right)}^{2}}\right]$$

$\bar{DWs}$ = average of all genotypes under stress conditions.

$\bar{DWc}$ = average of all genotypes biomass under control conditions.

DWs = average of individual genotypes biomass under stress.

DWc = average of individual genotype biomass under control conditions.

* Leaf relative water content (RWC) was determined according to the methods of Barrs and Weatherley [62] as follows:$$RWC=100\times \left[\frac{FW-DW}{SW-DW}\right]$$
where FW is the fresh weight of the leaves, DW is the dry weight of the leaves after 3 days of drying at 70 °C, and SW is the turgid weight of the leaves after 4 h of soaking in water at room temperature (approximately 20 °C).

* Relative osmolyte content (ROC) was calculated according to the Boyle–Van’t Hoff equation [63]:$$ROC=\Psi w\times \left[\frac{RWC}{RgT}\right]$$
where Ψw (MPa) is the water potential, R_g_ is the ideal gas constant (8.314 J K^−1^ mol), T (K) is the temperature in the psychrometer chamber, and ROC (mol g^−1^) is the relative osmolyte content.

* Relative soil moisture (RSM): The wet weight of the pots was measured daily by a balance with an accuracy of 0.01 g. By using it and the dry soil weight measured at the beginning of the experiment, the daily RSM can be calculated as follows:$$RSM(\%)=100\times \frac{\theta }{\theta f}$$
where θ is the weight of the water content of the soil; θf is the field water holding capacity.
$$\theta =100\times \left[\frac{Ww-Wd}{Wd}\right]$$
where Ww is the soil wet weight; and Wd is the soil dry weight.

* Potassium use efficiency for plant growth (KUE-DW) was calculated as the ratio of total plant biomass (g DW) to shoot K concentration (µg g^−1^ DW).

* Potassium use efficiency for photosynthesis (KUE-An) was calculated as the ratio of net photosynthesis (mol CO_2_ m^−2^ S^−1^) to shoot K concentration (µg g^−1^ DW).

* Water use efficiency for plant growth (WUE-DW) was calculated as the ratio of total plant biomass (g DW) to leaf water content (mL g^−1^ DW).

* Water use efficiency for photosynthesis (WUE-An) was calculated as the ratio of net photosynthesis (mol CO_2_ m^−2^ S^−1^) to leaf water content (mL g^−1^ DW).

## 5. Statistical Analysis

The software StatPlus Pro was used to analyze data and statistics. All data are presented as the mean ± standard deviation error. An analysis of variance (ANOVA) was performed to determine whether the differences between factors are significant. Fisher’s least significant difference test (LSD) at 5% was used. Significance was assigned when the difference between any two treatments was greater than the LSD value generated from the ANOVA. They are marked by different letters in the figures.

## Figures and Tables

**Figure 1 plants-12-01420-f001:**
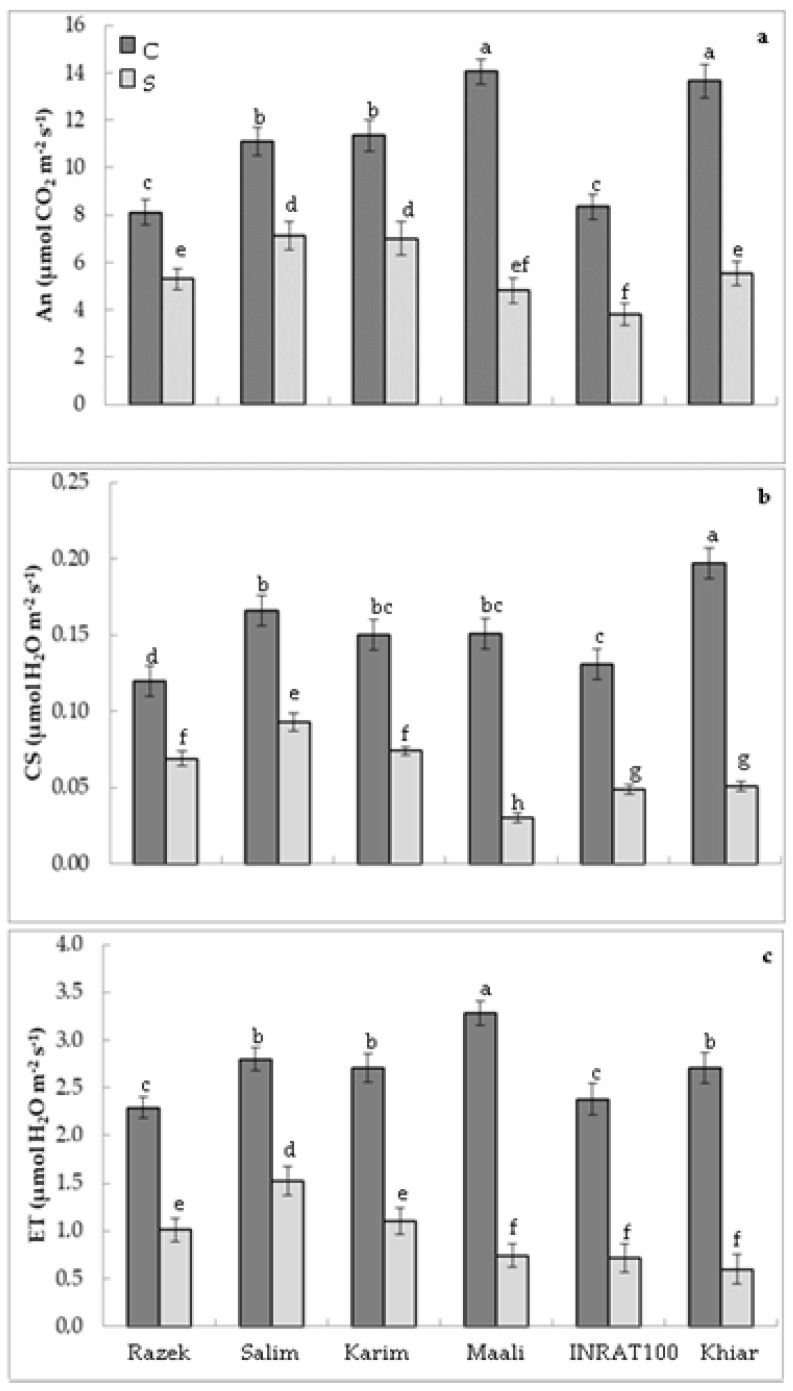
(**a**) Net photosynthetic assimilation (An), (**b**) stomatal conductance (SC), and (**c**) evapotranspiration (ET) in durum wheat (*Triticum durum* Desf.) plants subjected (S, stressed) or not (C, control) to drought stress. According to Fisher’s least significant difference, within columns, means with the same letter are not significantly different at α = 0.05. Standard errors of means of 30 replicates.

**Figure 2 plants-12-01420-f002:**
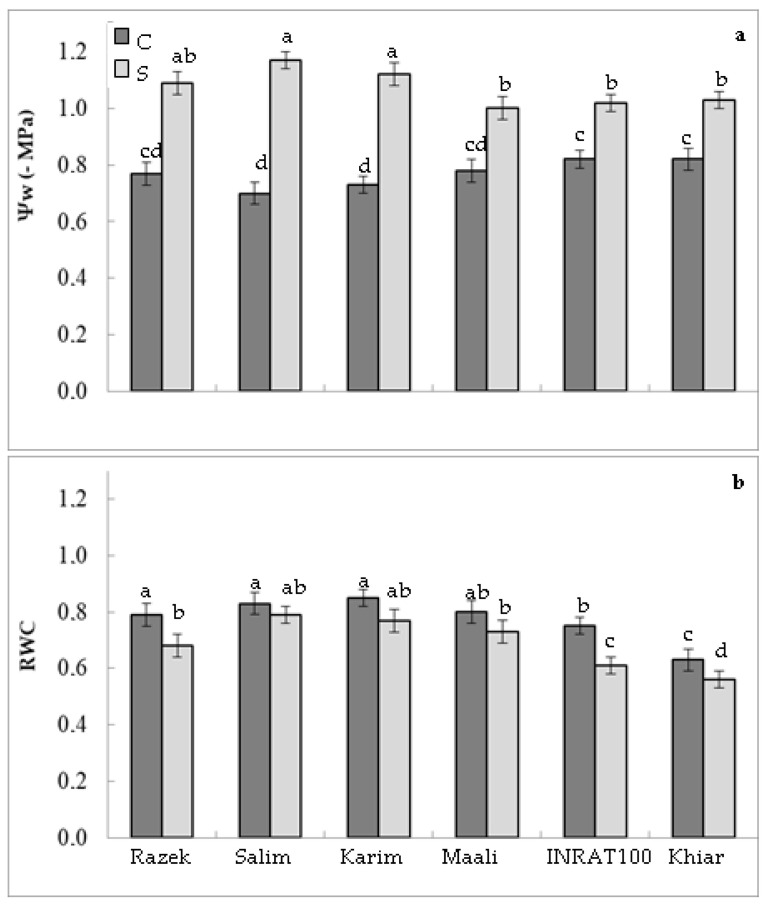
Water potential (Ψw, (**a**)) and relative water content (**b**) in durum wheat (*Triticum durum* Desf.) plants subjected (S, stressed) or not (C, control) to drought stress. According to Fisher’s least significant difference, within columns, means with the same letter are not significantly different at α = 0.05. Standard errors of means of 10 replicates.

**Figure 3 plants-12-01420-f003:**
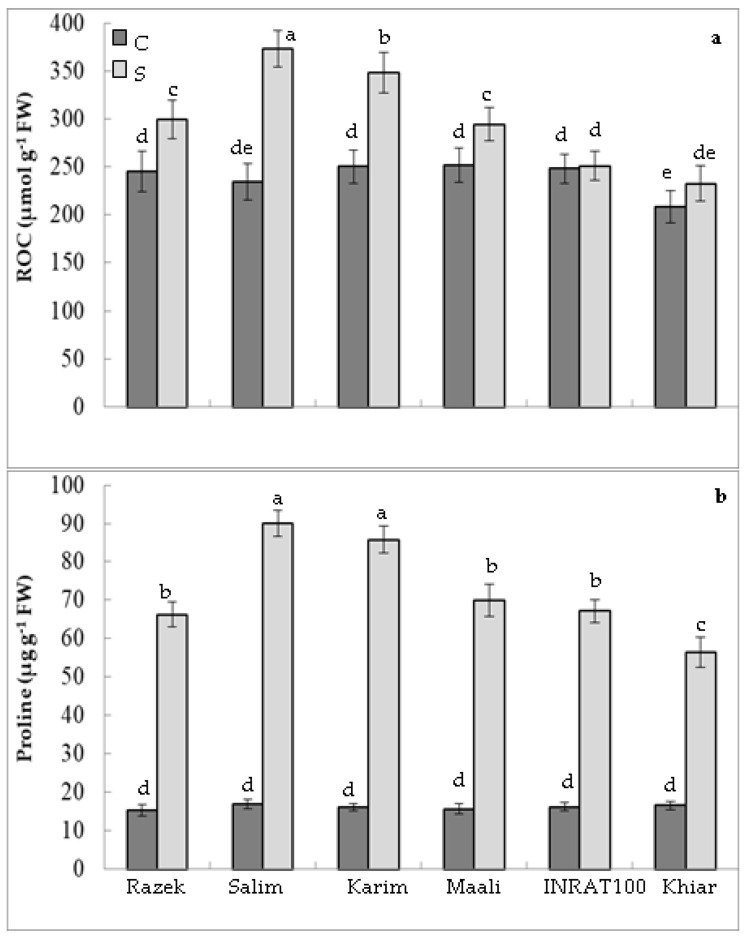
Relative osmolytes content (ROC, (**a**)) and proline concentration (**b**) in of durum wheat (*Triticum durum* Desf.) plants subjected (S, stressed) or not (C, control) to drought stress. According to Fisher’s least significant difference, within columns, means with the same letter are not significantly different at α = 0.05. Standard errors of means of 10 replicates.

**Figure 4 plants-12-01420-f004:**
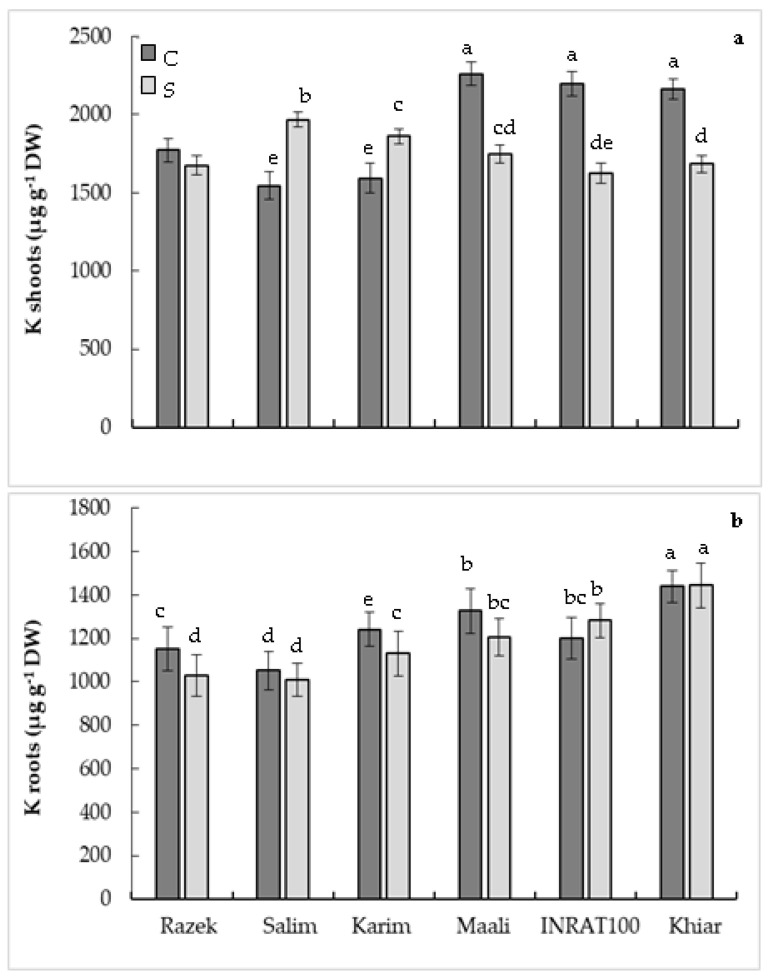
Potassium concentration in shoots (**a**) and roots (**b**) of durum wheat (*Triticum durum* Desf.) plants subjected (S, stressed) or not (C, control) to drought stress. According to Fisher’s least significant difference, within columns, means with the same letter are not significantly different at α = 0.05. Standard errors of means of 10 replicates.

**Figure 5 plants-12-01420-f005:**
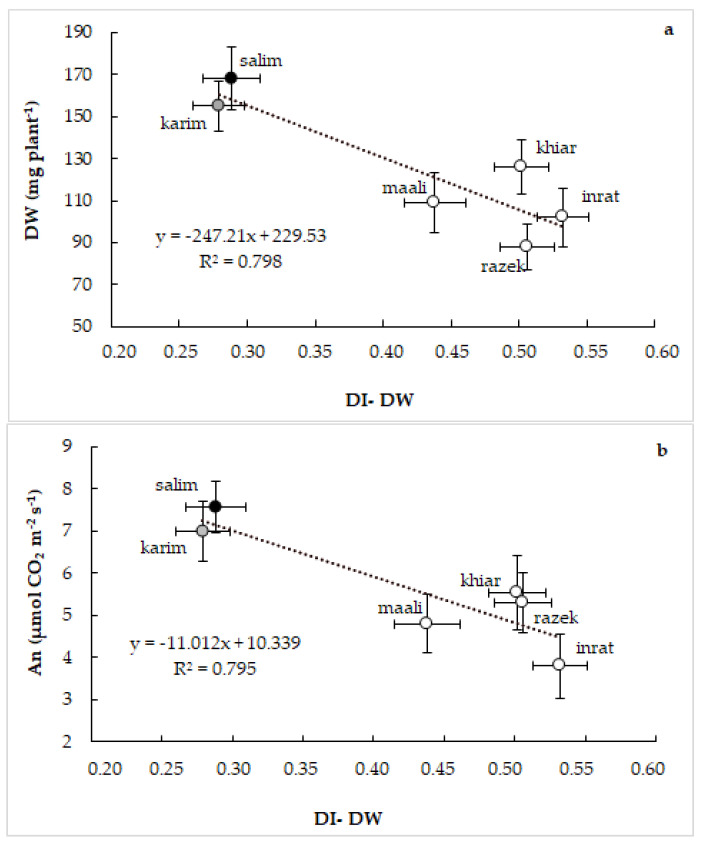
Relationship between plant growth (DW) and drought index based on dry weight (DI-DW, (**a**)), and between net photosynthesis (An) and drought index based on dry weight (DI-DW, (**b**)) in six Tunisian genotypes of durum wheat (*Triticum durum* Desf.) subjected to drought stress. Vertical and horizontal standard errors of means of 30 replicates.

**Figure 6 plants-12-01420-f006:**
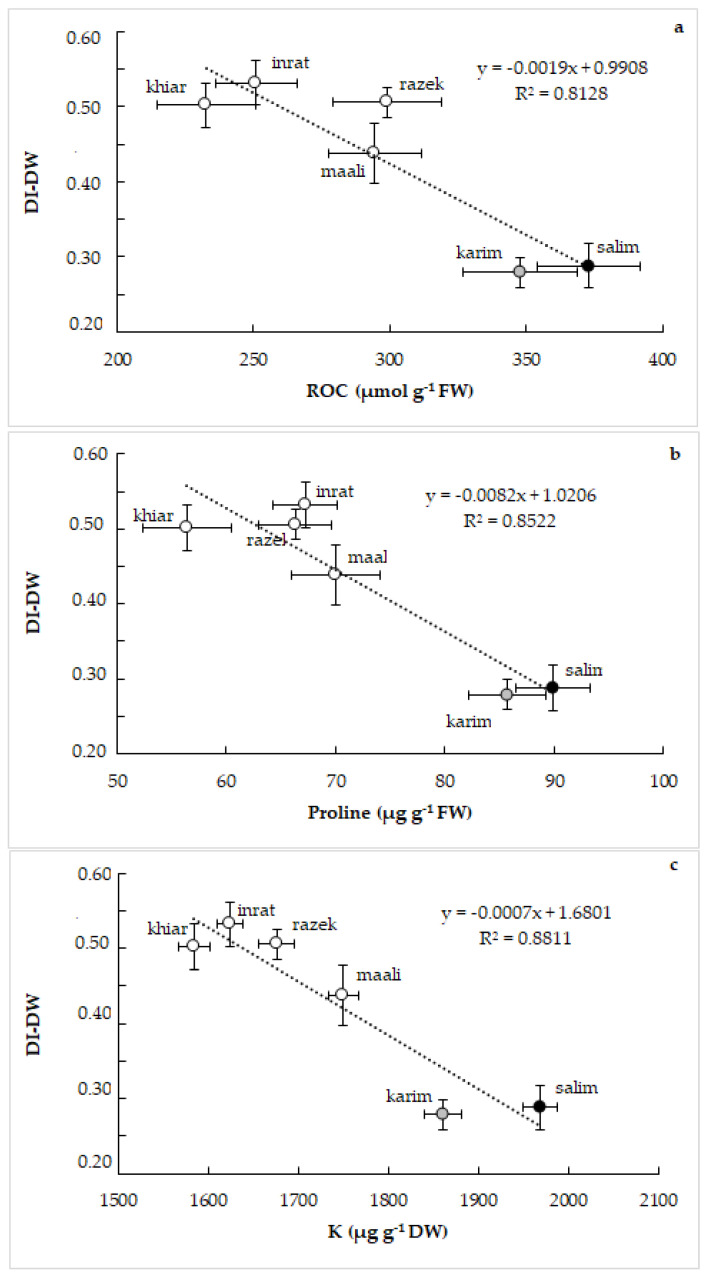
Relationships between drought index based on dry weight (DI-DW) and relative osmolyte content (ROC, (**a**)), between drought index based on dry weight (DI-DW) and proline concentration (**b**), and between drought index based on dry weight (DI-DW) and potassium (K) concentration (**c**) in shoots of durum wheat (*Triticum durum* Desf.) subjected to drought stress. Vertical standard errors of means of 30 replicates; horizontal standard errors of means of 10 replicates.

**Figure 7 plants-12-01420-f007:**
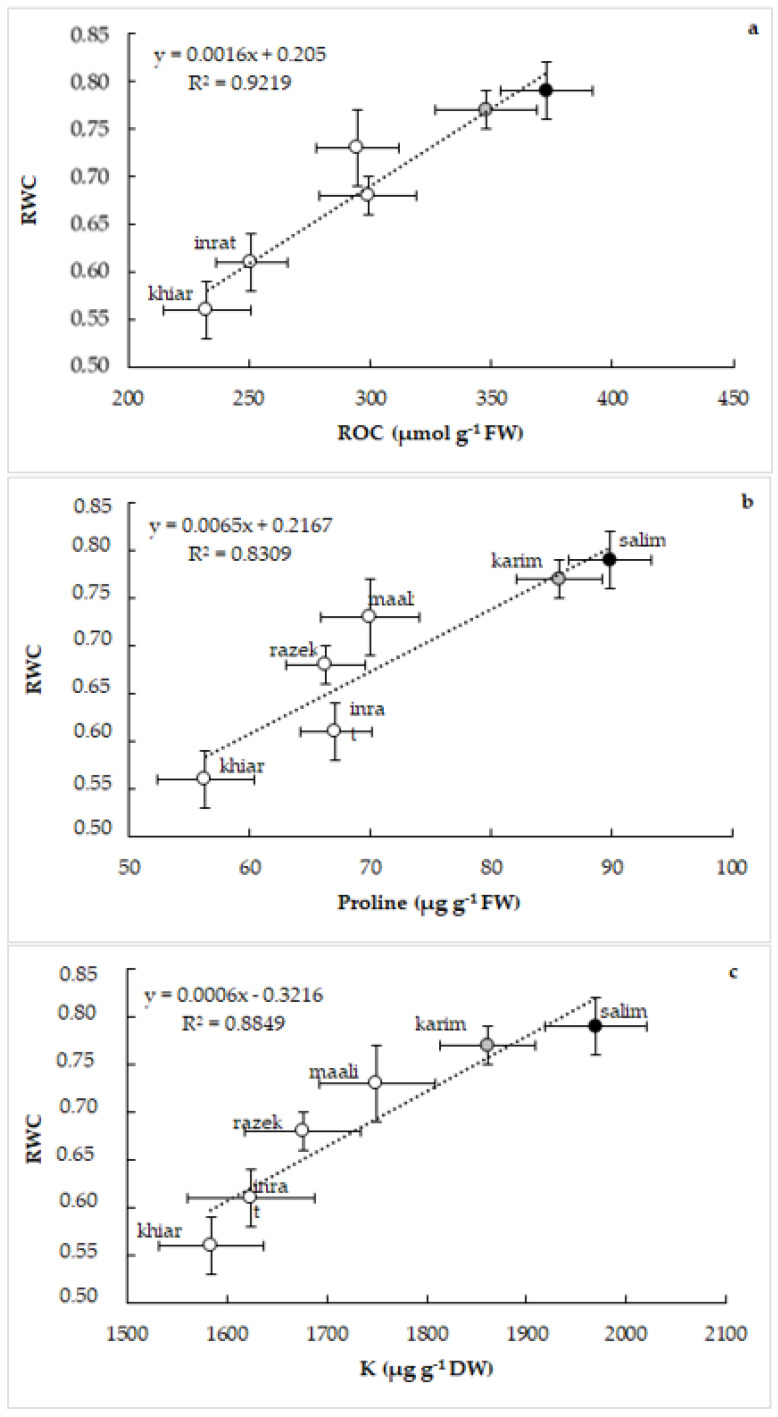
Relationships between relative water content (RWC) and relative osmolyte content (ROC, (**a**)), between relative water content (RWC) and proline concentration (**b**), and between relative water content (RWC) and potassium (K) concentration in shoots (**c**) of durum wheat (*Triticum durum* Desf.) subjected to drought stress. Vertical and horizontal standard errors of means of 10 replicates.

**Figure 8 plants-12-01420-f008:**
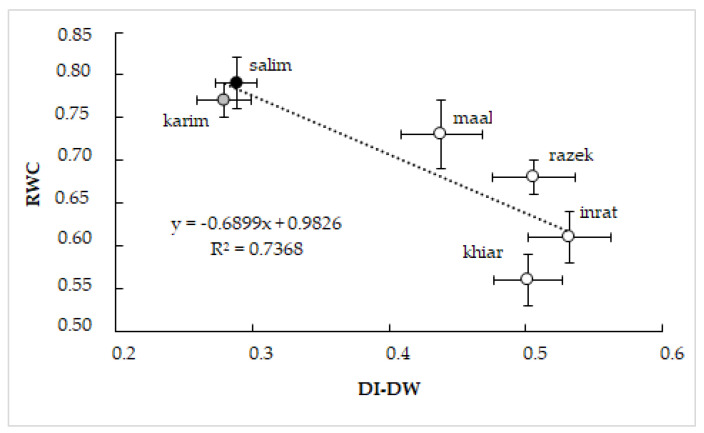
Relationship between relative water content (RWC) and drought index based on dry weight (DI-DW) in durum wheat (*Triticum durum* Desf.) genotypes subjected to drought stress. Vertical standard errors of means of 10 replicates; horizontal standard errors of means of 30 replicates.

**Figure 9 plants-12-01420-f009:**
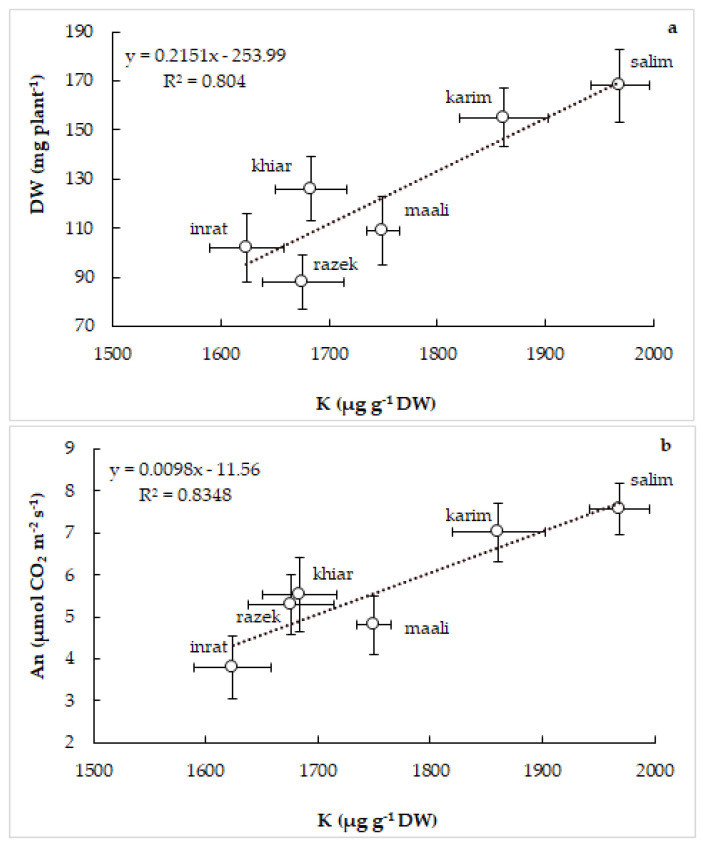
Relationship between biomass production (DW) and shoot K concentration (**a**) and between net photosynthesis (An) and shoot K concentration (**b**) in durum wheat (*Triticum durum* Desf.) plants subjected to drought stress. Vertical standard errors of means of 30 replicates; horizontal standard errors of means of 10 replicates.

**Table 1 plants-12-01420-t001:** Spad index, dry biomass production (DW), drought index calculated based on Spad index (DI-Spad), biomass production (DI-DW), and net photosynthesis (DI-An), and ratio of k shoots to K roots in six Tunisian genotypes of durum wheat (*Triticum durum* Desf.) subjected to drought stress. According to Fisher’s least significant difference, within columns, means with the same letter are not significantly different at α = 0.05. Standard errors of means of 30 replicates (10 replicates for K analysis). C: control, S: stressed.

	Spad Index		DW (mg Plant^−1^)		DI			K Shoots/K Roots	
	C	S	C	S	Spad	DW	An	C	S
razek	30.08 ± 1.8 ^d^	18.05 ± 1.17 ^f^	178 ± 15.2 ^e^	88 ^k^ ± 6	0.40 ^ab^ ± 0.032	0.51 ^b^ ± 0.038	0.35 ^de^ ± 0.029	1.5 ^bc^ ± 0.10	1.6 ^bc^ ± 0.13
salim	42.52 ± 1.57 ^b^	30.52 ± 2.51 ^d^	236 ± 16.4 ^b^	168 ^f^ ± 15	0.28 ^c^ ± 0.019	0.29 ^c^ ± 0.019	0.36 ^d^ ± 0.031	1.5 ^bc^ ± 0.11	2.0 ^a^ ± 0.15
karim	42.20 ± 2.21 ^b^	31.28 ± 3.20 ^d^	215 ± 16.6 ^c^	155 ^g^ ± 12	0.26 ^c^ ± 0.021	0.28 ^c^ ± 0.021	0.38 ^d^ ± 0.028	1.3 ^c^ ± 0.09	1.6 ^bc^ ± 0.11
maali	45.13 ± 1.42 ^a^	25.27 ± 0.79 ^e^	194 ± 15.3 ^d^	109 ^i^ ± 14	0.44 ^a^ ± 0.033	0.44 ^b^ ± 0.34	0.66 ^a^ ± 0.042	1.7 ^b^ ± 0.12	1.5 ^bc^ ± 0.09
inrat	39.22 ± 1.40 ^c^	25.50 ± 1.51 ^e^	218 ± 17.1 ^c^	102 ^j^ ± 11	0.35 ^b^ ± 0.027	0.53 ^a^ ± 0.041	0.54 ^c^ ± 0.037	1.8 ^ab^ ± 0.11	1.3 ^c^ ± 0.08
khiar	44.73 ± 1.01 ^a^	26.33 ± 0.71 ^e^	253 ± 16.2 ^a^	126 ^h^ ± 10	0.41 ^ab^ ± 0.036	0.50 ^ab^ ± 0.043	0.59 ^b^ ± 0.044	1.5 ^bc^ ± 0.12	1.2 ^c^ ± 0.08

**Table 2 plants-12-01420-t002:** Water use efficiency for plant growth (WUE-DW), water use efficiency for photosynthesis (WUE-An), potassium use efficiency for plant growth (KUE-DW), and potassium use efficiency for photosynthesis (KUE-An) in six Tunisian genotypes of durum wheat (*Triticum durum* Desf.) subjected (S, stressed) or not (C, control) to drought stress. According to Fisher’s least significant difference, within columns, means with the same letter are not significantly different at α = 0.05. Standard errors of means of 10 replicates.

	WUE-DW		WUE-An		KUE-DW		KUE-An	
	C	S	C	S	C	S	C	S
razek	2.17 ^d^ ± 0.17	1.22 ^f^ ± 0.11	0.10 ^b^ ± 0.011	0.07 ^b^ ± 0.005	100.46 ^d^ ± 8.3	52.506 ^h^ ± 4.2	4.58 ^c^ ± 0.41	3.16 ^e^ ± 0.21
salim	3.06 ^a^ ± 0.21	2.27 ^a^ ± 0.18	0.14 ^ab^ ± 0.012	0.10 ^a^ ± 0.010	152.74 ^a^ ± 10.2	85.331 ^e^ ± 6.3	7.18 ^a^ ± 0.43	3.84 ^d^ ± 0.23
karim	2.79 ^b^ ± 0.18	2.18 ^b^ ± 0.20	0.15 ^a^ ± 0.011	0.10 ^a^ ± 0.010	134.88 ^b^ ± 11.1	83.289 ^e^ ± 6.6	7.12 ^a^ ± 0.52	3.77 ^d^ ± 0.22
maali	2.37 ^c^ ± 0.19	1.70 ^d^ ± 0.15	0.17 ^a^ ± 0.013	0.08 ^ab^ ± 0.006	85.84 ^e^ ± 6.7	62.286 ^g^ ± 5.1	6.22 ^b^ ± 0.47	2.74 ^f^ ± 0.19
inrat	2.63 ^bc^ ± 0.20	1.55 ^e^ ± 0.12	0.10 ^b^ ± 0.010	0.06 ^bc^ ± 0.004	99.11 ^d^ ± 8.5	62.808 ^g^ ± 5.1	3.79 ^d^ ± 0.24	2.34 ^g^ ± 0.17
khiar	2.98 ^a^ ± 0.21	1.88 ^c^ ± 0.14	0.16 ^a^ ± 0.012	0.08 ^ab^ ± 0.006	116.98 ^c^ ± 9.1	74.822 ^f^ ± 5.6	6.31 ^b^ ± 0.46	3.29 ^e^ ± 0.24

**Table 3 plants-12-01420-t003:** Drought susceptible index (DSI) was calculated based on the Spad index (DSI-Spad), biomass production (DSI-DW), and net photosynthesis (DSI-An), and the drought tolerance index was calculated based on the Spad index (DTI-Spad), biomass production (DTI-DW), and net photosynthesis (DTI-An) in six Tunisian genotypes of durum wheat (*Triticum durum* Desf.) subjected to drought stress. According to Fisher’s least significant difference, within columns, means with the same letter are not significantly different at α = 0.05. Standard errors of means of 10 replicates.

	DSI-Spad	DSI-DW	DSI-An	DTI-Spad	DTI-DW	DTI-An
razek	1.22 ^a^ ± 0.091	1.20 ^b^ ± 0.11	0.83 ^d^ ± 0.051	0.33 ^d^ ± 0.021	0.34 ^d^ ± 0.025	0.60 ^d^ ± 0.046
salim	0.79 ^d^ ± 0.052	0.68 ^d^ ± 0.049	0.65 ^f^ ± 0.044	0.79 ^a^ ± 0.046	0.85 ^a^ ± 0.061	1.28 ^a^ ± 0.11
karim	0.73 ^d^ ± 0.048	0.66 ^d^ ± 0.048	0.75 ^e^ ± 0.047	0.80 ^a^ ± 0.055	0.72 ^b^ ± 0.049	1.24 ^a^ ± 0.093
maali	1.13 ^b^ ± 0.088	0.92 ^c^ ± 0.087	1.35 ^a^ ± 0.12	0.69 ^b^ ± 0.047	0.50 ^c^ ± 0.031	1.03 ^c^ ± 0.085
inrat	0.98 ^c^ ± 0.067	1.26 ^a^ ± 0.099	1.29 ^b^ ± 0.091	0.61 ^c^ ± 0.043	0.48 ^c^ ± 0.033	0.60 ^d^ ± 0.031
khiar	1.15 ^b^ ± 0.11	1.19 ^b^ ± 0.077	1.21 ^c^ ± 0.098	0.71 ^b^ ± 0.053	0.69 ^b^ ± 0.047	1.15 ^b^ ± 0.077

## Data Availability

Not applicable.

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
