# Peer review of "Functional Dissection of the Physiological Traits Promoting Durum Wheat (Triticum durum Desf.) Tolerance to Drought Stress"

_plants, 2023, doi:10.3390/plants12071420_

Round 1

Reviewer 1 Report

Screening and identification of wheat varieties for drought tolerance is a very routine work. In this study, only 6 varieties were selected for variety screening, but the sample size was too small, and there was no explanation as to why these 6 varieties were selected and what was the particularity of these 6 varieties?

Therefore, this study can only be regarded as the initial part of the research work. Simple physiological data cannot explore the nature of the differences in drought tolerance of the six varieties, and the research work is not innovative.

Author Response

* The reason for choosing 6 genotypes was explained in introduction section (final paragraph that highlight the problematics and objectives)

Durum wheat is a strategic crop in Tunisia, cultivated in winter and harvested in summer. It occupies most of the agricultural land in the North where rainfall usually meets the water needs of plants. The main critical stage of the plant cycle is the vegetative development that influences the reproductive stage (flowering, ear setting and ear filling). However, most of the research on drought stress has been done at the grain filling stage and little the works that gave an interest to the vegetative stage. Furthermore, climate change has pushed aridity to spread over almost all of Tunisian land and durum wheat fields (with specific rain-fed genotypes) that were in humid climates now belong to the semi-arid or arid climate. Their response to water scarcity must therefore be studied. Most of the research on drought stress in wheat is done at the grain filling stage, also named terminal drought or postanthesis drought [28]. Exploring the genotypic variability of response to drought stress among wheat genotypes allows us to screen tolerant ones and identify the associated useful traits of tolerance. Accordingly, this study was planned and consists of subjecting six Tunisian genotypes to water stress (karim, khiar, inrat, maali, razek, salim). They are widely cultivated in the north of the country where the climate is ordinary humid, while karim crops extend to the center (semi-arid climate).

* The sample size was too small

We are sorry for the mistake made in the figure legends. In fact, the results presented are the means of 10 repetitions, but per plot (missing sentence), so means of 30 repetitions for biomass production and all nondestructive measurements (spad, photosynthethesis and associated calculation, DI….). Just for chemical and biochemical analysis results are presented as means of 10 replicates (RWC, K, proline and ROC).

Corrections are made in figures and tables legends and in MM section.

* To improve the data for the exploration of differences in drought tolerance of the six genotypes,

we added some available results from the same experiment that have not been used in the first version of the MS. Proline and relative osmolyte content are thus added.

Reviewer 2 Report

Comments to MS “Functional dissection of the physiological traits promoting Durum wheat (Triticum durum Desf.) tolerance to drought stress” written by Salim Ltaief and Abdelmajid Krouma

This manuscript focuses on some physiological parameters in foliar organs of 6 genotypes of  Triticum durum exposed to drought. The analysis of the role of different plant organs and especially leaves on physiology of plants and development was described in many other papers, and it has its economic meaning. In ’80 and ‘90 it was demonstrated many times that leaves contribute to build plant, fruits and seeds biomass. We learned many details about the role of these organs, however, our knowledge about photosynthesis of leaves and Triticum crops is rather scant. In addition  there have been few studies up to now of the biochemical consequences and of the parallel changes in photochemistry in plant organs other than leaves. This may be expected to be investigated during next decades. It is really interesting subject in plant physiology of agricultural plants. The problem in this manuscript is clearly stated but the general intention is extremely simple. There is huge number of issues Author should address before this manuscript could be considered acceptable.

Major criticism: Author should take the time to present a clear rationale and hypothesis what is the reason for testing just the role these organs of chosen genotypes. What is new for plant physiology or agriculture from presented research? What about photosynthesis in stems, petiols and midribs in non-foliar plant organs? Author should also write down the meaning of presented relationships for crop production. The data presented here are repetition of known results but in some places they are patchy and far from complete. From this MS we learn known dependencies about the physiology of some leaves but interpretation of presented facts is not clear. Discussion should point out the novel aspects of presented results for plant physiology and I hope it is possible to draw some interesting general conclusions on the base of presented references. Thus, the data presented here are simple repetition and they need adequate interpretation.

Abstract: In this paper results of some shown measurements are presented without general conclusion. Presented results give some indications on crops development in other plants and this would be interesting for potential Plants reader. When writing about photosynthesis it would be necessary to know the role of photosynthesis in chosen leaves building crops. This would help to draw conclusions and to know about origin of CO2 in plant body. When we use in experiments such important crop this would be nice to see what is the effect on productivity.

Key words:

It is not necessary to repeat words used in the title.

Introduction and Discussion

The discussion contains a lot of speculations and could be shortened by weeding out speculative statements. Several statement are typical examples of overinterpretation. I hope some speculations can be replaced with some other very precise conclusions.

The results shown in this paper could be in future also useful for practice. Presented results will be not interesting to other plant scientists as this report on known  changes in plant metabolism described in thousands other papers and explored in other publications.

Some information on productivity of foliar organs/tissues would be interesting when such information is available.   

This MS should point out the novel aspects of presented results for physiology of plants and/or indicate if this can help in cultivation of used genotypes of wheat. There are some interesting observations which should be indicated and discussed. The meaning of presented results should be discussed in terms of donor-sink possible metabolite exchange in plants exposed to stresses.

It is not like this that amount of chlorophyll correlate with intensity of photosynthesis. This is much more complicated. Higher concentration of chlorophyll does not mean stimulation of photosynthesis. 

You should decide to write Potassium” or “potassium” , “K” or “k”, “Figure” or “figure”.

M&M

When writing “fully expanded leaf” is not precise which one did you use for experiments.

Generally, the paper is NOT concisely written. Despite the above criticism I would say that here presented experimental results after correction and carefully discussion can be interesting for potential Plants reader.

I hope that my criticism and suggestions may help the Author to achive the goal and that above comments will be of value to Author. 

Author Response

The overall manuscript was revised, and a lot of improvements are made in all sections (Abstract, introduction, MM, results and discussion). They are highlighted in green. To justify the problematic and objectives of the study, the last paragraph in introduction was rewritten, to consolidate and confirm obtained results and justify the observed genotypic differences, some new date from the same experiment were added (relative osmolyte content, proline), results and discussion are thus readjusted

Reviewer 3 Report

The findings demonstrate the interdependence of osmotic fluctuations, photosynthetic activity, and plant development. The authors showed that the ability of Durum wheat to endure (or be sensitive to) water stress considerably influences its potential to retain better leaf hydration. They also concluded that the capacity of water to influence photosynthesis and plant growth is significant. Besides, they suggested that uptake and buildup of potassium in leaves are the main elements of this mechanism. They considered the genotypes Salim and Karim are able to withstand drought stress because of a healthy equilibrium between a number of physiological parameters, including stomatal conductance, photosynthesis, transpiration, water and potassium utilization efficiency, and transpiration. In further screening processes, the physiological characteristics of drought tolerance can be used for KUE, WUE, DI, DSI and DTI can be used to arrange the studied cultivars according to their tolerance.

The comments are attached in the file 

Author Response

Thank you very much for the comments and the attached file.

All comments are retained, and appropriate improvements are made

Reviewer 4 Report

Dear Authors,

Your paper entitled “Functional dissection of the physiological traits promoting Durum wheat (Triticum durum Desf.) tolerance to drought stress” to determine six Tunisian wheat genotypes tolerance to drought stress. The manuscript is too wordy and the structure of the most sentences is poor and even is not complete. It is hard to follow the entire manuscript due to the length of the manuscript. This manuscript must be re- written in a clear format with coherent sentences with clear objective for each paragraph and/or sentences.  The manuscript has to be easy to digest by the reader and easy to follow.

Specific comments

Line 14: provide the scientific name for wheat and where mentioned first in the induction.

Line 19: Can the authors explain what they are trying to deliver in this statement? “significantly reduced their water potential”

Line 20: Define K. Do not expect every reader knows what K stand for.

Line 28: provide reference “Over the past few decades, the intensity and extension of drought swept all regions of the globe.”

Line 36: “with 771 million tons per year that satisfy the demand of 21% of the 36 world’s population.” Modify it to with 771 million tons production per year which satisfies the demand of 21% of the 36 world’s population.

Author Response

Thank you very much for accepting to review our MS and for the recommendations you presented.

The MS was revised and re-adjusted to be more comprehensive, some results are added to improve date for analysis of the differential response to drought stress among the studied genotypes. Problematics and objectives are more clarified in the last paragraph of introduction, and MM improved by new amendments, results and discussion are also re-adjusted. Thank you for the specific comments that have been taken into consideration.

Round 2

Reviewer 1 Report

Authors introduced some small changes, however the most important weak points are not removed. From physiological point of view this MS can be treated as repetition of well known results. Presented data can be interested for breaders but not for scientists.

Author Response

Thant you very much for your comments

Our MS underwent a total overhaul a first and second time. The scientific content has been improved by the addition of new data (proline and relative osmolytes) as well as the introduction and discussion have been taken up and improved. I suppose that the scientific content of the manuscript is largely satisfactory given the large number of physiological and biochemical parameters developed and the quality of the results obtained. In this MS we identified some traits of durum wheat tolerance to water stress by linking plant growth, key metabolic functions (photosynthesis and chlorophyll), osmotic adjustment with its organic component (proline...) and its inorganic component (K) and water relations (RWC, Ψw). This study allowed us to identify the common thread between all these parameters and functions. This study allowed us to identify the common thread between all these metabolic parameters and functions and identify the traits that modulate genotypic differences in durum wheat response to drought stress. Scientifically, this article is very rich and represents a functional dissection of durum wheat variability in response to water stress involving key metabolic functions.

Please find attached the new version of MS in which the first improvement are highlighted in green and the second highlighted in blue.

Best regards

Reviewer 2 Report

No significant improvement of this MS was done. All described relationships are published many times in other papers. 

Author Response

Dear Professor

Thank you very much for the consistent review of our MS.

Our MS was the subject of a deep revision. All sections are revised regarding their scientific content, paragraph coherence and references. Abstract, introduction and discussion are clearly improved, sufficient information and actual references were added. All amendments added after the 1st and 2nd round of revision were highlighted in green.

  1. To provide sufficient background and include all relevant references, the introduction was almost rewritten and clearly improved. The introduction is reorganized according to a more scientific approach, starting with an overview on water stress as a major global problem, then durum wheat as a strategic and vital agricultural product. The following paragraph was reserved for the adaptive mechanisms of plants to water stress. Thus, morpho-physiological and biochemical mechanisms were approached with reference always to new works and results. The problem, the choice of genotypes and thebjecttive of this work aroused a primary interest in the last paragraph of the introduction.
  1. References cited were improved by adding others more actual (2022-2023) and the list rose to 64 instead of 40 in the first version and 55 after the first round of review. Convenient references were added in the paragraph that seemed weak.
  2. The conclusion was rewritten and clearly improved. The main findings are highlighted and consistently presented.

With best regards

Pr. Krouma Abdelmajid

Reviewer 4 Report

Comments to the authors,

The manuscript has improved after the authors incorporated the comments and suggestions. However, the manuscript is still wordy (long and difficult to follow) please eliminate the unnecessary sentences and the relative words and information.  It must be coherent and informative with less words than this.

Specific comments:

Modify to Figure 1. (a) Net photosynthetic assimilation (An), (b) Stomatal conductance (SC)  and (c) evapotranspiration (ET)  in Durum wheat (Triticum durum) plants subjected (S, stressed) or no (C, control) to drought stress. According to Fisher's Least Significant Difference, within columns, means with the same letter are not significantly different at α = 0.05. Standard errors of means of 30 replicates. Do the same for the rest of the figures. 

define all the abbreviations that were mentioned in the title of the figures and tables.

Author Response

Thank you very much for accepting to review our MS and for the recommendations you presented.

Point 1. The manuscript has improved after the authors incorporated the comments and suggestions. However, the manuscript is still wordy (long and difficult to follow) please eliminate the unnecessary sentences and the relative words and information.  It must be coherent and informative with less words than this.

The manuscript was taken up and subjected to a thorough revision and reading. Thus an improvement has been made, especially in the discussion section in order to lighten the content and better appear the message. Some unnecessary sentences and words have been eliminated and others have been rewritten. They are highlighted in blue in the MS

Point 2. Specific comments:

Modify to Figure 1. (a) Net photosynthetic assimilation (An), (b) Stomatal conductance (SC)  and (c) evapotranspiration (ET)  in Durum wheat (Triticum durum) plants subjected (S, stressed) or no (C, control) to drought stress. According to Fisher's Least Significant Difference, within columns, means with the same letter are not significantly different at α = 0.05. Standard errors of means of 30 replicates. Do the same for the rest of the figures. 

The recommended modifications have been made. They are highlighted in blue in the MS

Point 3. Define all the abbreviations that were mentioned in the title of the figures and tables.

Figures and tables titles are revised, all necessary abbreviation are defined. They are highlighted in blue in the MS

With best regards
